# Osteoneogenesis at the Round Window: A Possible Cause of Cochlear Implant Failure?

**Giulia Donati, Nader Nassif and Luca Oscar Redaelli de Zinis ***

Department of Medical and Surgical Specialties, Radiological Sciences and Public Health, Section of Audiology, University of Brescia, 25100 Brescia, Italy; ju.donati@hotmail.it (G.D.); nadernassif@alice.it (N.N.)
* Correspondence: luca.redaellidezinis@unibs.it; Tel.: +39-0303996236

**Abstract:** Surgery for cochlear implant is a traumatic procedure, with inflammatory responses leading to immediate and delayed intracochlear changes, resulting in newly formed fibrous and bony tissue. This newly formed tissue is thought to affect speech perception with cochlear implants and can also play a role in causing device malfunctioning and soft failures. We present a case of left cochlear implant explantation and reimplantation in a 15-year-old girl, who experienced deterioration of speech perception and device failure associated with osteoneogenesis of the round window, which could represent a cause of cochlear implant failure. To avoid surgical trauma of the cochlear lateral wall, enlarged round window insertion rather than a cochleostomy, soft surgical techniques, and the application of steroids are all important issues to prevent new tissue formation, although special attention should also be given to the trauma of round window borders.

**Keywords:** osteoneogenesis; round window; cochlear implant; device failure

## 1. Introduction

Even though revision surgery for cochlear implantation is an unusual occurrence, recent studies report a reimplantation rate in children between 3% and 18% [1–5]. Most authors agree that the pediatric population demonstrates a higher incidence compared to the adult population, probably because of an increased incidence of head trauma, higher risk of middle ear infection, and electrode migration associated with the normal growth of the skull or other factors [6].

The indications for cochlear implant (CI) surgical revision include two main types: device-related and non-device-related. Device-related failures are divided into hard and soft failures.

The former is a loose connection between the speech processor and the internal device; a typical hard failure in children is direct damage to the receiver stimulator due to head trauma, which represents 56% of hard failures and is due to motor vehicle collisions, all-terrain vehicle accidents, and falls [5]. The most common manufacturer's analysis finding in the case of hard failure is electrode array damage, followed by loss of hermetic seal, receiver stimulator damage, circuit abnormality, electronic damage/malfunction, and normal analysis [7].

Soft failures are those that occur when the connection is preserved with declining performance and symptoms without a clear identification of the cause, such as performance issues (sudden drop or slower decrement in hearing over time or intermittent performance), audiologic symptoms (atypical tinnitus, buzzing, and popping sounds), non-auditory symptoms (pain at the implant site, facial stimulation, rejection of speech processor use), and mapping problems (changes in pulse width or duration, no response in electrically evoked compound action potential measurements, high impedances or open circuits on telemetry, 'no communication with implant' error, implant coupling problems, significant changes in comfortable levels, decrease in number of active electrodes) [8,9]. A recent

example is that of the recall of two models by Advanced Bionics (AB, Valencia, CA) stemming from low electrode impedance values and hearing performance declines for some users [10]. Some reports suggested that an immunologic response to the electrode with a potential local reaction might explain other CI soft failures [11,12].

Non-device-related or medical causes include device infection or extrusion, inadequate initial placement, surgical wound or flap complications, electrode array migration (idiopathic fibrosis and ossification of the basal turn or skull growth in these cases may cause extrusion of electrode), internal package migration, head trauma that causes a seroma or hematoma with possible infection and biofilm formation, hematoma, cerebral-spinal fluid leakage, and technological upgrade [6]. Other medical causes are strictly related to middle ear pathologies, including chronic otitis media, retraction pockets, and iatrogenic or residual cholesteatoma [13].

New bone formation at the lips of the round window (RW) has never been associated with CI failure, and the observation of extended ossification at the lips of the RW in a patient with CI failure after 6 years from implantation prompted us to report the event and discuss how to prevent it.

## 2. Case Presentation

A 15-year-old girl was referred for left cochlear reimplantation for device failure. The girl had been followed by our audiological department from the age of 4 when she came from India. Her parents were not relatives and were of normal hearing. Bilateral severe hearing loss of unknown origin was diagnosed in her country at the age of 2, and hearing aids were prescribed. In our audiological department, at the age of 5, following one year follow-up with hearing aids, it was decided to implant the right ear and maintain the hearing aid on the left side. The array was introduced through an anterior inferior promontorial cochleostomy. Four years later, the left ear received no benefit from the hearing aid, and it was decided to also implant the left ear. She received a left Cochlear© contour advance implant through a RW access, lowering the lips of the RW. Five years after left implantation, the speech processor showed no connection to the implant and the patient reported a mild worsening of hearing threshold, though this was not affecting speech comprehension. An integrity test was performed by Cochlear Italia SRL. The receiver stimulator test indicated intermittent output, and the CI was classified as "characteristics decrement (B1)" according to the "International Classification of Reliability for Implanted Cochlear Implant Receiver Stimulators" [14]. In the following months, the patient experienced a progressive and significant decline in her hearing with inadequate clinical benefit; thus, the result from the integrity test was reclassified as "device failure (C)". Reimplantation surgery was proposed, but the patient decided to be reimplanted only 18 months later. Intraoperatively, the iuxtafenestral portion of the array electrode was compressed by enveloping and newly formed bone tissue (Figure 1).

Drilling was necessary for array mobilization and explantation and for optimal visualization of the RW (Figure 2).

The new electrode array (Cochlear© contour advance implant) was completely inserted through an anteriorly and inferiorly enlarged RW access without any obstacle (Figure 3) and was surrounded by a large amount of connective tissue to prevent damage by potential new bone formation (Figure 4).

The NRT threshold was normal in all the electrodes. Analysis of the explanted implant performed by the Device Analysis Department (Cochlear Limited, Sydney, Australia) found damage of the electrode array and hermetic seal failure, and we cannot exclude that the damage detected was not related to the explant maneuvers. The new CI was activated 1 month after surgery. The girl always showed good performances with both her left and right CI and in binaural conditions. At the last follow up, 16 years after right-sided surgery and 6 years after left reimplantation, the left CI ensured aided thresholds in the 25 to 30 dB HL range. Scores with both CIs together were 100% in vowel identification, 70% in consonant identification, 100% in words recognition, in open-set sentences recognition, and

100% in questions comprehension, which were improved compared with the best results obtained before left CI failure (90% vowel identification, 70% consonant identification, 90% words recognition, 90% question comprehension). Free field vocal audiometry showed 90% word recognition at 55 dB. The matrix sentence test speech reception threshold was 9.8 dB signal to noise ratio. Datalogging showed 7 h/day of bilateral use.

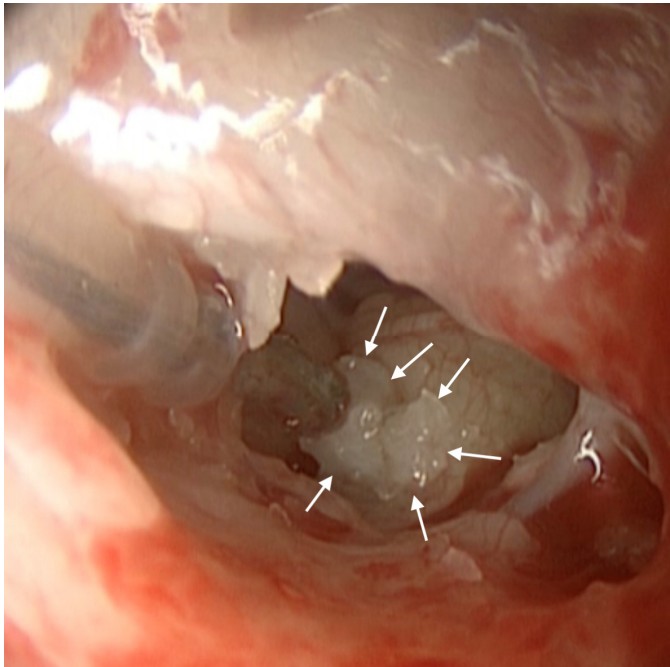

**Figure 1.** Posterior tympanotomy exposure of the region of the RW: white arrows indicate the new bone formation around the array.

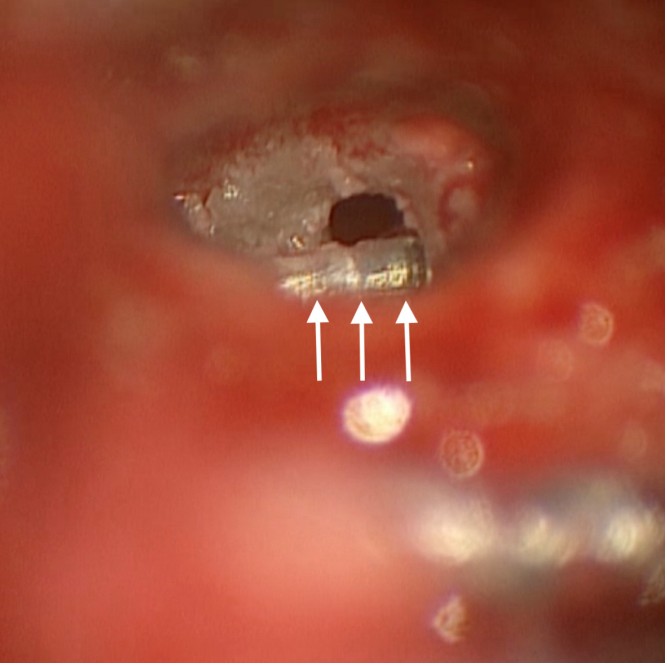

**Figure 2.** Posterior tympanotomy exposure of the region of the RW: white arrows indicate the portion of the bent array that was enveloped by the new bone removed.

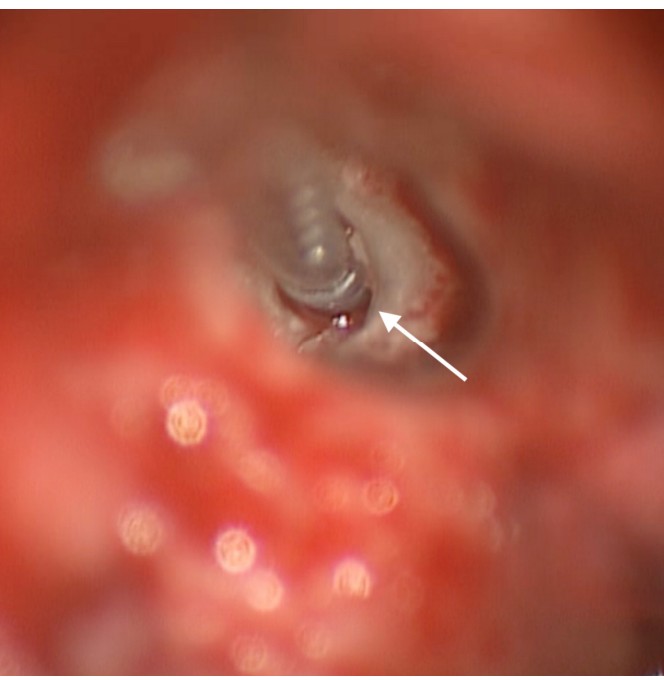

**Figure 3.** Posterior tympanotomy exposure of the region of the RW: white arrow indicates the enlarged RW and the complete insertion of the array.

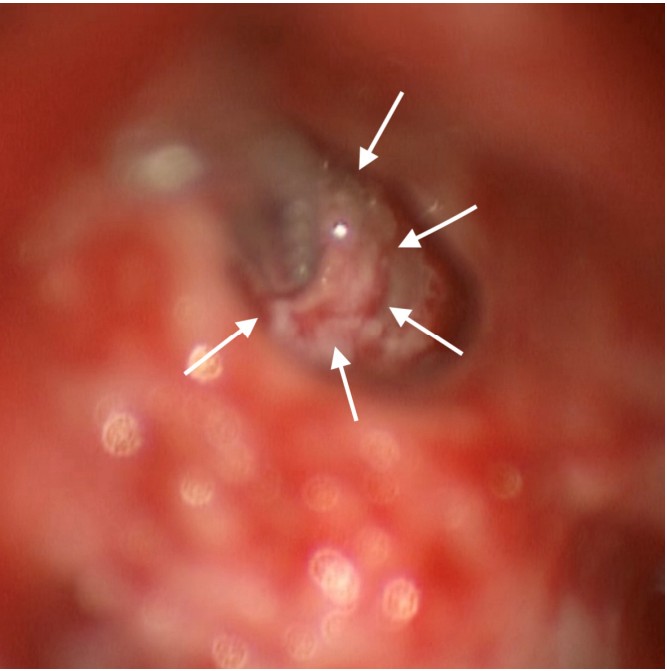

**Figure 4.** Posterior tympanotomy exposure of the region of the RW: white arrows indicate the connective tissue positioned between the border of the enlarged RW opening and the array.

Written informed consent to present the case was obtained from the patient and her parents.

### 3. Discussion

New bone formation at the cochleostomy site, but also along the implantation track to a variable length, and sometimes until the end of the implant array, has been clearly

documented in the literature and almost always reported in revision cases or in postmortem analyses of temporal bones with a CI [15–18].

Factors which may induce cochlear bone and fibrous tissue neoformation after cochlear implantation are not clearly known. The most credited hypothesis is surgical trauma. Immediate intracochlear changes to surgical injury are due to an acute inflammatory response to the array insertion [19] and include disruption of the spiral lamina, interruption of the basilar membrane, damage of the spiral ligament, stria vascularis, lateral cochlear wall, and modiolus [17,18,20,21]. Delayed damages occur because of both insertional trauma and foreign body reaction of the host to the electrode itself [19].

Severe insertional trauma to the lateral cochlear wall by an electrode may both expose the endosteum and provide a focal point of inflammation to promote ossification [17]. The inflammatory mediators may diffuse throughout the cochlear fluid space and contribute to a general increase in new tissue formation, as opposed to a localized area of new tissue formation adjacent to the trauma [17]. This may explain why those authors did not find an association between localized damage to the lateral cochlear wall and localized new tissue formation. An alternative explanation they gave is that the spiral ligament may have a regulatory function in the metabolism of the bone in the otic capsule [17].

It is well demonstrated that inflammatory cytokines play a role in cochlear ossification after meningitis [22]. Likewise, the decrease in complement, a mediator of inflammation, reduces the extent of cochlear fibrosis and labyrinthitis ossificans after inducted meningitis in Mongolian gerbils [23]. This hypothesis for a cellular inflammatory response has become evident through human temporal bones studies from CI recipients [17,18,21]. The cellular inflammatory response and the amount of ossification are greatest at the cochleostomy and in the basal turn and decrease towards the cochlear apex [17–19].

A trend for increasing new bone formation with time after implantation has been hypothesized [17,18,24], even though some authors have not found any correlation [19,25,26].

Other elements such as age at implantation, size, site of cochleostomy, and electrode design are believed to affect this biologic response [18].

It is of uncertain significance if the presence of this newly formed tissue can affect CI functional performances since the results in the literature are conflicting. According to Kawano et al. [20], the hearing thresholds for single electrodes increased with greater volumes of new bone and fibrous tissue, whereas the dynamic ranges were decreased, meaning that, potentially, these modifications could negatively affect speech perception. Similarly, other studies found an inverse correlation between the amount of new tissue formation and CI speech performances [19,24].

The lack of a significant correlation found by other authors [17,18,21,27] could mean that intracochlear new tissue is not an important determinant of performance, or that there is limited power to notice a consequence, due to the small sample size and the different otologic history of patients.

Intracochlear new bone and fibrous tissue may have other adverse consequences: compromising reimplantation surgery, alterations in the transmission of the current, obliteration of the scala tympani, or the destruction of hair cells, which all affect residual hearing [17,18]. One additional consequence of bone regrowth is electrode migration: intracochlear ossification can move the electrode array out of the cochlea [28].

Our intraoperative evidence of bone regrowth in the RW region is consistent with the traumatic cause of bone regrowth, whereas the damages reported in the analysis of the explanted implant in the present case cannot be conclusive on the cause of CI failure. The localization of massive bone regrowth corresponded to site of trauma due to drilling of the hook area. The absence of excessive fibrous tissue or bone regrowth within the scala tympani, confirmed by the easy removal of the damaged array and complete introduction of the electrode array, is evidence that there was little insertion trauma in the previous operation. Most cases can be reimplanted through the same route of the first implantation, but RW reimplantation has also been reported after first implant through a promontorial

cochleostomy [29], and the RW approach was converted to cochleostomy due to new bone formation and granulation tissue at the insertion site [9].

Given a causal relation between insertional distress to the lateral wall and consequent fibro-ossification, the use of perimodiolar electrodes, RW insertion rather than a lateral wall cochleostomy, soft surgical techniques, and the application of steroids have been suggested to minimize endosteum damage and reduce the subsequent tissue reaction [17,24,25,30]. Regarding our observation of a large amount of new bone formation at the cochleostomy site possibly damaging the electrode, we enlarged the RW access (Figure 3) under continuous irrigation and enveloped the electrode by thick connective tissue (Figure 4) during revision surgery. This preventive measure could also be used during primary intervention to preserve cochlea and CI functioning over time since the number of young patients implanted with residual hearing is progressively increasing [26]. Furthermore, an enlarged RW access allows one to choose the better trajectory of insertion, avoiding the luminal wall and modiolus trauma, and allows for the perilymph to come out from the cochlea during electrode insertion, decreasing intracochlear pressure so that the chances of residual hearing preservation increase [31]. In children, reduced visibility of the RW niche from the posterior tympanotomy approach is frequently reported [32]. In these cases, the assistance of a 1.9 diameter rigid endoscope through the posterior tympanotomy allows for the complete visualization of the RW and the direction of the first curve of the basal turn of the cochlea after opening the RW. This can help in axis determination of the proper insertion of the electrode to reduce the trauma of insertion [32].

In our patient, 6-year post-reimplant audiological performance with both CIs together were improved compared with the best results obtained before left CI failure (100% vs. 90% vowel identification, 70% vs. 70% consonant identification, 100% vs. 90% words recognition, 100% vs. 90% question comprehension), even though there was a late reimplantation (18 months after device failure).

Even though most reimplanted patients reach at least the same performance than before the failure [5,33], there are still 15–16% of children with worsening performances [1,34]. Different possible causes of deteriorating performances have been suggested: scala vestibuli insertion of the electrode array, initial diagnostic error, soft tissue collapse occurring in the scala tympani impeding correct reinsertion, long periods of non-use or limited CI usage prior to reimplantation, and poor adherence to the speech rehabilitation program after reimplantation [1,34].

Explanations for improved performance after reimplantation have also been suggested: soft surgery in revision, fibrous sheath induced by the primary implantation guides reinsertion of the electrode array without extra trauma, improved design of new electrodes, technical progress of new-generation implants, greater number of electrodes, improved acoustic discrimination, maturation and brain development in children, and reactivation and recovering of development under auditory stimulation of the auditory areas and nerve, which were previously under-stimulated by the defective implant [35]. Possible immediate worsening in the activation after reimplanting in children has been reported, with successive and stable improvement until at least 10 years after reimplantation [36]. Substantial favorable results of reimplantation should be considered by the CI team when counselling with patients and parents to motivate on the prospects of improved performance with reimplantation [5]. Continuous monitoring of patient performance allows us to promptly identify decreases in functionality of the first CI. The knowledge of prognostic factors is important to help patients in new rehabilitation, keeping in mind that possible initial worsening will be followed by probable improvement with strict adherence to the rehabilitation program [34].

## 4. Conclusions

The need for cochlear reimplantation is to be kept in consideration by physicians, particularly for children. Improved performances are realizable provided that proper new rehabilitation is followed. Osteoneogenesis of the RW could represent a rare source of CI

failure that can be treated by reimplantation. The soft and slow insertion of the array into the scala tympani through an enlarged RW approach and the application of steroids can reduce surgical trauma of the cochlear lateral wall to prevent subsequent intracochlear new bone formation. Special attention should also be given in preparing the RW to visualize the proper trajectory for array introduction. The surrounding electrode array at RW access by a sufficient amount of connective tissue can prevent implant damage by potential new bone formation.

**Author Contributions:** Conceptualization, L.O.R.d.Z.; surgical intervention, L.O.R.d.Z.; data curation, L.O.R.d.Z.; writing—original draft preparation, G.D.; writing—review and editing, N.N.; supervision, L.O.R.d.Z. All authors have read and agreed to the published version of the manuscript.

**Funding:** This research received no external funding.

**Institutional Review Board Statement:** The study was conducted in accordance with the Declaration of Helsinki and approved by the Ethics Committee of ASST Spedali Civili Brescia (protocol code 3754, 24 September 2019).

**Informed Consent Statement:** Informed consent was obtained from the subject involved in the study.

**Data Availability Statement:** Data are unavailable due to privacy.

**Conflicts of Interest:** The authors declare no conflicts of interest.

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
