# Peer review of "Osteoneogenesis at the Round Window: A Possible Cause of Cochlear Implant Failure?"

_audiolres, doi:10.3390/audiolres14010001_

Round 1

Reviewer 1 Report

Comments and Suggestions for Authors

Th authors have very clearly shown a case of CI delayed failure.

At the surgical revision they found a new hypothetical cause

They are showing the way to run out the case and avoid reappearance of osteoneogenesis surrounding the implant

I just miss some dat of the first (right) ear implanted. Whether or not a same procedure was taken and why the situation occurs just  in the second one.

Author Response

The authors have very clearly shown a case of CI delayed failure. At the surgical revision they found a new hypothetical cause They are showing the way to run out the case and avoid reappearance of osteoneogenesis surrounding the implant. I just miss some data of the first (right) ear implanted. Whether or not a same procedure was taken and why the situation occurs just in the second one.

We detailed the description of the case to answer to the question of the reviewer: in the first ear the array was inserted through an anterior inferior promontorial cochleostomy. In the first intervention for the second ear the lips of the round window niche were lowered but the insertion was incomplete, so in the revision the round window was enlarged anteriorly and inferiorly to identify the correct direction of the scala tympani.

Reviewer 2 Report

Comments and Suggestions for Authors

The authors present a case report of  cochlear implant failure due to osterogenesis of the round window. It is an interesting topic, but the literature is rather poor. However, there is a number of issues that need to be addressed, prior to consideration for publication.

• The abstract is well structured.

• The introduction provides a proper background, but it could probably be upgraded with additional more references.

• Concerning the case presentation please provide some information regarding the general medical history and family history of the patient, or any other examinations that possibly were performed to detect any systematic diseases.

• Figures from the surgical field are useful for the comprehension of the operation. Since some of the are pretty blurred, it would be nice if the quality of the definition could be enhanced.

• In the discussion part the authors  try to support their assumptions with notable references but it could be upgraded with some more recent articles that are encountered in the literature.

We look forward to your revision.

Comments on the Quality of English Language

The quality of English language is quite good. A review by a native English speaking editor/service check for a number of minor but significant grammatical/syntax issues would be appreciated.

Author Response

The authors present a case report of cochlear implant failure due to osterogenesis of the round window. It is an interesting topic, but the literature is rather poor. However, there is a number of issues that need to be addressed, prior to consideration for publication.The abstract is well structured.

The introduction provides a proper background, but it could probably be upgraded with additional more references.

We performed a new literature search (previous one was in September 2023) and upgraded the introduction.

Concerning the case presentation please provide some information regarding the general medical history and family history of the patient, or any other examinations that possibly were performed to detect any systematic diseases.

We specified that there were no hearing problems in the parents and in other relatives in the case presentation.

Figures from the surgical field are useful for the comprehension of the operation. Since some of the are pretty blurred, it would be nice if the quality of the definition could be enhanced.

We are sorry but blurred vision cannot be eliminated by enhancing definition because it is a consequence of the limited depth of field in high magnification pictures. The arrows show the focused parts which are important for comprehension. 

In the discussion part the authors try to support their assumptions with notable references but it could be upgraded with some more recent articles that are encountered in the literature.

We upgraded the discussion and added a part related to the audiological results of reimplantations.

Revised manuscript was checked by an English native speaker proofreader.

Reviewer 3 Report

Comments and Suggestions for Authors

The presented case is a particular one because most intraoperative descriptions refer to ossification of the cochlea, ossification around the electrode array, or the cochleostomy, and not at the RW level.

Regarding the case, its description should be more detailed. We do not know what was the cause of the hearing loss what treatments the patient had before the implantation, why such a long interval was expected until the implantation of the 2nd ear and what was the postoperative evolution. It would be important to know if there were more special events, in time, that could have explained the osteoneogenesis, apart from the surgical trauma.

In my opinion the conclusions are a little bit excessive. It is recommended to use the perimodiolar electrode because the straight one would damage the endosteum and cause ossification, which is not the case in this presentation.

Author Response

The presented case is a particular one because most intraoperative descriptions refer to ossification of the cochlea, ossification around the electrode array, or the cochleostomy, and not at the RW level. Regarding the case, its description should be more detailed. We do not know what was the cause of the hearing loss what treatments the patient had before the implantation, why such a long interval was expected until the implantation of the 2nd ear and what was the postoperative evolution. It would be important to know if there were more special events, in time, that could have explained the osteoneogenesis, apart from the surgical trauma.

As required, we detailed the description of the case to answer as well as possible to the questions of the reviewer.

In my opinion the conclusions are a little bit excessive. It is recommended to use the perimodiolar electrode because the straight one would damage the endosteum and cause ossification, which is not the case in this presentation.

We remove the part of the conclusion with the recommendation to use the perimodiolar electrode enhancing the observations gained by our experience

Round 2

Reviewer 2 Report

Comments and Suggestions for Authors

thank you for your revision